# Systemic Antibiotics and Chlorhexidine Associated with Periodontal Therapy: Microbiological Effect on Intraoral Surfaces and Saliva

**DOI:** 10.3390/antibiotics12050847

**Published:** 2023-05-04

**Authors:** Stella de Noronha Campos Mendes, Camila Machado Esteves, Juliana Alethusa Velloso Mendes, Magda Feres, Nathalia Figueiredo, Tamires Szeremeske de Miranda, Jamil Awad Shibli, Luciene Cristina Figueiredo

**Affiliations:** 1Department of Periodontology, Dental Research Division, Guarulhos University, Guarulhos 07090-023, SP, Braziljualethusa@hotmail.com (J.A.V.M.); mferes@ung.br (M.F.);; 2Department of Dentistry, Federal University of Piaui, Teresina 64049-550, PI, Brazil; 3Department of Oral Medicine, Infection, and Immunity, Division of Periodontology, Harvard School of Dental Medicine, Boston, MA 02115, USA

**Keywords:** periodontal diseases, biofilms, saliva, tongue, antibacterial agents, antibiotics, metronidazole

## Abstract

The effect of systemic antibiotics on the microbial profile of extracrevicular sites after periodontal treatment is currently the subject of research. This study evaluated the microbiological effects on different oral cavity sites of scaling and root planing (SRP) combined with antimicrobial chemical control in the treatment of periodontitis. Sixty subjects were randomly assigned to receive SRP alone or combined with metronidazole (MTZ) + amoxicillin (AMX) for 14 days, with or without chlorhexidine mouth rinse (CHX) for 60 days. Microbiological samples were evaluated by checkerboard DNA–DNA hybridization until 180 days post therapy. The adjunctive use of antibiotics plus CHX significantly reduced the mean proportions of red complex species from subgingival biofilm and saliva (*p* < 0.05). Furthermore, the analysis of all intraoral niches showed a significantly lower mean proportion of the red complex species in the same group. In conclusion, the concomitant use of antimicrobial chemical control (systemic and local) demonstrated a beneficial effect on the composition of the oral microbiota.

## 1. Introduction

Scaling and root planing (SRP) have been the most common forms of periodontal therapy, and their clinical and microbiological effects are well documented [1,2]. Usually, this procedure improves clinical parameters, effectively reduces the probing depth, and promotes gain in clinical attachment level, especially at deeper sites [1,3]. However, it does not provide long-term stability of these clinical benefits [4].

The association of systemic and local chemical controls of dental biofilm with SRP has been suggested as an adjunct treatment for periodontitis. In this context, metronidazole (MTZ) combined with amoxicillin (AMX) appears to be the most favorable systemic antibiotic used in the treatment of periodontitis [4,5,6,7,8]. MTZ is a synthetic drug derived from nitroimidazole and presents bacterial and protozoan activity. MTZ acts on the bacterial cells by passive diffusion and provides toxic metabolites that interact with DNA and other bacterial macromolecules, causing cellular death [9]. However, this effect is limited to strictly anaerobic bacteria, such as *Porphyromonas gingivalis*, *Treponema denticola*, and *Tanerella forsythia*. AMX is a bactericidal antibiotic from the penicillin group that acts on anaerobic facultative bacteria, coccus, and Gram-negative and Gram-positive bacillus. AMX avoids the synthesis of the bacterial cell wall, resulting in bacterial death.

An earlier review [10] showed that scaling and root planning (SRP) associated with systemic MTZ + AMX reduced deeper pockets > 5 mm more effectively than SRP alone. Broad rebiosis in subgingival bacterial plaque goes beyond its effects in controlling species such as *Aggregatibacter actinomycetemcomitans*. MTZ + AMX acted on the reduction in several periodontal pathogens from the red and orange complexes and some newly identified taxa, with an increase in the proportions of host-compatible species [6,7,8,11,12]. In addition, combining chlorhexidine (CHX) mouth rinse with SRP leads to clinical benefits and can result in the reduction in periodontal pathogens from different intraoral surfaces and saliva [8,9,10,11,12,13,14].

The clinical success of periodontal treatment is achieved when the levels, proportions, and percentages of sites colonized by different periodontal pathogens are effectively reduced after therapy. Furthermore, a successful outcome is associated with a new microbial community with higher proportions of host-compatible microorganisms established in the subgingival biofilm [13,14,15]. Nevertheless, it has been demonstrated that these periodontal pathogens also colonize the tonsils, tongue, saliva, and oral mucous membranes. Therefore, it has been suggested that these areas might interfere in the recolonization of recently scaled pockets, thereby serving as reservoirs for the reinfection of the periodontium [11,15,16,17].

To the best of our knowledge, no other study has evaluated the changes in the microbial profile of soft tissues and saliva after periodontal treatment with systemic and local antimicrobials in subjects with periodontitis. Thus, this study evaluated the hypothesis that SRP combined with systemic MTZ and AMX associated with CHX mouth rinse would affect (H1) (or not (H0)) the microbiological composition of different oral cavity sites in the treatment of periodontitis.

## 2. Results

The study was conducted without dropouts during the course of the experimental period. All subjects reported full adherence to the prescribed course of antibiotic or placebo treatment and CHX rinses. One subject from the control group, one from the T2 group, and two from the T3 group reported some adverse events (diarrhea) and taste changes, but no statistically significant differences were observed between the groups.

Table 1 presents the subjects’ demographic and clinical data at baseline and 180 days after therapy. No statistically significant differences were observed among the four groups at baseline; however, the treatments differed substantially in terms of clinical outcome. The two groups that received adjunctive MTZ + AMX presented more significant reductions in the number of sites, with PD ≥ 5 mm, compared to subjects that received SRP alone or were associated with CHX (*p* < 0.05).

No significant differences were observed in the mean counts and proportions of the 40 bacterial species at baseline for any of the intraoral sites or saliva (*p* > 0.05). The four groups were microbiologically homogeneous at the beginning of the study. Figure 1, Figure 2, Figure 3, Figure 4 and Figure 5 present the mean counts (×10^5^) of the 40 species evaluated throughout the study for all selected niches. The species were grouped according to the microbial complexes described by Socransky et al. (1998) [18]. In general, the counts of most of the species found in the tongue coating, soft tissue, and supragingival biofilm did not change significantly from baseline to 180 days (Figure 1, Figure 2 and Figure 3). In saliva, subjects from the T3 group presented a significant increase in two Actinomyces species (*Actinomyces isralelli* and *Actinomyces naeslundii*) and a significant reduction in two red complex species (*Tanerella forsythia* and *Porphyromonas gingivalis*). In this group, there was also a significant reduction in the three red complex species and six orange complex species from subgingival biofilm (Figure 4 and Figure 5).

Figure 6 shows the changes in the proportions of the microbial complexes for all intraoral niches (saliva, tongue coating, soft tissue, supragingival, and subgingival biofilms) analyzed together in the four treatment groups during the experimental period. The microbial profiles were positively affected by treatments, and the most beneficial changes were observed in the T3 group (MTZ + AMX + CHX). These subjects showed a significant reduction in the proportion of red and orange complexes from baseline to 180 days post therapy and an increase in the proportion of *Actinomyces* sp. They also demonstrated a significantly lower mean proportion of red complex species than the other groups.

## 3. Discussion

In this investigation, similarly to the literature, the two groups that used SRP with adjunctive systemic antibiotics showed better clinical and microbiological effects than the group administered SRP alone. In addition, periodontal treatment with systemic antibiotics showed more significant reductions in the number of sites with PD ≥ 5 mm and reduced microbiological dysbiosis in subgingival plaque and saliva compared to mechanical debridement alone.

The protocol that associates SRP with MTZ and AMX has been strengthened since the publication of six systematic reviews on the topic [19,20,21,22,23,24] and several randomized clinical trials (RCTs) showing the clinical benefits of this therapy [25,26,27,28,29,30].

In addition, it has been shown that oral bacteria demonstrate specific tropisms toward the different biological surfaces in the oral cavity, such as the teeth, mucosa, and other bacteria. This is due to the different ecological characteristics of various oral environments [31]. However, while most studies have observed the microbiota of the teeth, which comprises only about 20% of the total area of the oral cavity, the role played by the composition of the soft tissue microbiota in periodontal diseases and their treatments has not received the same attention [31,32].

With respect to the composition of the microbiota of different environments in the mouth, the results of this investigation are in agreement with those of other studies that found detectable levels of periodontal pathogens in these habitats. For example, Dahlen et al. (1992) [33] examined the tongue coating of periodontally diseased and non-diseased young adult Kenyan subjects for seven putative periodontal pathogens. All test species, including *P. gingivalis*, *Prevotella intermedia*, *Campylobacter rectus*, and *Aggregatibacter actinonomycetemcomitans*, could be detected in samples from both groups of subjects. In addition, Mager et al. (2003) [31] evaluated microbiological samples of 8 oral soft tissue surfaces, as well as from saliva from 225 periodontally healthy subjects, using the checkerboard DNA–DNA hybridization technique and observed that all sites had different microbial colonization profiles and that periodontal pathogens were detected in all sites, especially the three red complex species.

Few studies have analyzed the effect of periodontal therapy on the microbiota of sites other than those in the subgingival environment [30,34,35]. Danser et al. (1996) [34] observed the effect of SRP and periodontal surgery on the levels of *A.actinomycetemcomitans*, *P. gingivalis*, and *P. intermedia* on the oral mucous membranes using indirect immunofluorescence. They found no reduction in the prevalence of the tested species after treatment. In addition, Quirynen et al. (1998) [35] examined the effect of a one-stage, full-mouth disinfection procedure on microbiological colonization of the tongue in periodontitis patients. The authors observed a reduction in the CFU/mL of black-pigmented species; however, the total bacterial levels were unchanged. To the best of our knowledge, this is the first study to evaluate the effect of systemic antibiotics on the changes in the oral microbiota using the checkerboard DNA–DNA hybridization technique. Our results show better microbiological benefits for the adjunctive use of antibiotics plus CHX (MTZ + AMX + CHX), which reduced several periodontal pathogens. The main contributors to these findings were saliva and subgingival biofilm.

Recently, Lu et al. (2022) [30] investigated the microbial shift after SRP-treated periodontitis with or without adjunctive antibiotics. A 6-month pilot randomized controlled trial recruited 14 subjects with severe periodontitis, who were divided into a test group receiving full-mouth SRP with or without AMX (500 mg) and MTZ (200 mg) (t.i.d. 7d) and a control group. Clinical examination, collection of subgingival plaque and saliva, and blood tests were performed at baseline pretreatment, as well as three months and six months post treatment. The periodontal condition significantly improved in both groups; the test group showed a greater improvement in the plaque index, probing depth, and bleeding index than the control group. The test group demonstrated significantly lower microbial richness and diversity and less abundant Porphyromonas than the control group at three months for both the subgingival and salivary microbiome. However, the microbial differences narrowed within six months. The subgingival and salivary microbiota shifted synergistically.

It is interesting to consider that saliva harbors bacteria from different niches of the oral cavity, acting as a reservoir of microorganisms. After periodontal treatment, these bacteria can pass from saliva or soft tissues to other areas, recolonizing the subgingival region [36]. Thus, reducing the number of bacteria in the saliva is essential to minimize reinfection by subgingival pathogens. In this sense, antibiotics contribute to the control of infection in the subgingival environment and other oral niches, playing a synergistic role in this process. Therefore, comparing the microbiological profile of different oral sites, in addition to the supra- and subgingival biofilms, after periodontal treatment may help to understand the clinical and microbiological effects of periodontal therapy on the patterns of subgingival recolonization.

The main limitation of this study is the closed-end microbial test used, which precludes the identification of uncultivated organisms. Next-generation sequencing evaluation has increased the knowledge of bacterial communities, including as-yet-uncultured taxa [37]. Although studies using next-generation sequencing techniques are encouraged when conducting microbiome studies of oral health and disease, the 40 species of the checkerboard DNA–DNA hybridization panel continue to be consistent markers for oral dysbiosis and homeostasis, especially regarding microbiological outcomes of periodontal treatment. Furthermore, the checkerboard test allows for the quantification of individual species, contributing data not provided by sequencing techniques. Finally, studies using metatranscriptomic analysis to assess the metabolic functions and the virulence factors expressed by these potential novel pathogens would be fundamental to broadening the vision of the complex functionality of the oral microbiome. Therefore, future studies that focus on unveiling the oral microbiome present in several intraoral surfaces in both healthy and diseased states require the use of multiple sequencing techniques, including 16S rRNA gene sequencing, metagenomic sequencing, transcriptomics, proteomics, and metabolomics, through standardized, large, multicenter case–control studies.

## 4. Materials and Methods

### 4.1. Sample Size Calculation

The ideal sample size to ensure adequate power for the microbiological data of this study was calculated considering differences of at least 6.6 percentage points between groups for the proportion of the red complex species with a standard deviation of 5.0 percentage points [36]. These calculations determined that 15 subjects per group would be necessary to provide 85% power with an α of 0.05.

### 4.2. Subject Population and Inclusion/Exclusion Criteria

The study sample was composed of 60 subjects with stage II and III periodontitis (≥30 years) based on the International Classification of the American Academy of Periodontology (2018). Patients were selected from the population referred to the Periodontal Clinic of Guarulhos University (Guarulhos, SP, Brazil). The volunteers were chosen after the project’s evaluation and approval by the Research Ethics Committee of Guarulhos University (#15/2007). This study was conducted in full accordance with the World Medical Association Declaration of Helsinki. All participants were informed of the study’s objectives, risks, and benefits, as well as the types of clinical measurements. Participation in the survey was voluntary, and individuals or their caregivers signed a free and informed consent form in accordance with the Directives and Norms of the National Health Council. Participating individuals did not suffer any type of interference or change to their treatment plan.

The inclusion criteria were subjects who had not previously received periodontal therapy for at least 6 months, had at least 15 teeth (excluding third molars), and a minimum of 6 teeth with at least one site with a probing depth (PD) and clinical attachment level (CAL) between 5 and 10 mm. Exclusion criteria were as follows: smokers and ex-smokers of less than 5 years; pregnant or lactating; systemic diseases that compromise the host response or require prophylactic medication for treatment; prolonged use of anti-inflammatory or immunosuppressive medications (over 3 months) and antibiotics in the last 6 months prior to study participation; continuous use of oral antiseptics; allergy to CHX, MTZ, and/or penicillin; and extensive prosthetic rehabilitation.

### 4.3. Experimental Design

In this single-blinded, placebo-controlled study, subjects were randomly assigned to one of the following treatment groups using a computer-generated table:Control (C, n = 15): SRP + AMX placebo + MTZ placebo + CHX placebo;Test 1 (T1, n = 15): SRP + AMX placebo + MTZ placebo + CHX 0.12%;Test 2 (T2, n = 15): SRP + AMX (500 mg) + MTZ (400 mg) + CHX placebo;Test 3 (T3, n = 15): SRP + AMX (500 mg) + MTZ (400 mg) + CHX 0.12%.

All subjects received full-mouth supragingival scaling and were instructed on proper home-care techniques. The same dentifrice was given to all subjects to use during the study (Colgate Total^®^, Colgate Palmolive Co., São Bernardo do Campo, SP, Brazil). Subjects received full-mouth SRP performed under local anesthesia in a maximum of six appointments of 1 h each. Treatment of the entire oral cavity was completed in a maximum period of 14 days. SRP was manually performed by two trained periodontists. All subjects received maintenance therapy after 3 months.

After the first section of SRP, the antibiotic (AMX/MTZ) and placebos were administered three times a day for 14 days, and the supragingival biofilm control was achieved by rinsing with 15 mL of 0.12% CHX solution for 1 min twice a day for 60 days. The antibiotics, mouth rinses, and placebos were prepared by Pharmédica pharmacy (São Paulo, SP, Brazil). The placebos and their respective active substances (antibiotic or CHX) were identical (i.e., same taste and color) to their counterparts and were placed in identical bottles (antibiotic) and flasks (mouth rinses). The coded bottles were given to the examiners, who did not have access to information about the contents of the bottles or the assignment of subjects to therapies at any time during the study. During the visits, all patients answered a questionnaire about any self-perceived side effects of the medication/placebo, and examiners were responsible for calling the subjects every 2 days to monitor compliance.

### 4.4. Microbiological Monitoring

All subjects received microbiological monitoring at baseline, as well as 45, 90, and 180 days post therapy. The non-stimulated saliva samples were collected for 1 min in sterile tubes with glass pearls. The soft tissue samples were collected by gently stroking the buccal mucosa with a swab microbrush for approximately 30 s, searching for the adherence of microorganisms to the bristles. The coating was taken with only one movement from an area of 10 mm in length measured with a periodontal probe (North Carolina periodontal probe, Hu-Friedy, Chicago, IL, USA) and 5 mm in width, corresponding to the tip of a 11–12 Gracey curette (Hu-Friedy, Chicago, IL, USA) of the dorsum of the tongue from a region posterior to the sulcus terminals [37]. Supragingival and subgingival biofilm samples from sites with PD between 5 and 7 mm and CAL between 5 and 10 mm were collected from two sites (one superior and one inferior) per subject with individual sterile mini Gracey curettes (#11-12).

All samples from different sites (a sample of 0.03 mL saliva, tongue coating, one microbrush from soft tissue, supragingival, and subgingival biofilm) were immediately placed in separate Eppendorf tubes containing 0.15 mL of TE (10 mM Tris-HCl, 1 mM EDTA, pH 7.6-Sigma-Aldrich, Sao Paulo, SP, Brazil). One hundred microliters of 0.5 M NaOH (Sigma-Aldrich, Sao Paulo, SP, Brazil) was added to each tube, and the samples were dispersed using a vortex mixer. The tubes were stored at −20 °C until analysis by checkerboard DNA–DNA hybridization technique for 40 bacterial species. The entire microbiological analysis was performed at the Laboratory of Microbiology of Guarulhos University. The samples were boiled for 10 min and neutralized using 0.8 mL of 5 M ammonium acetate. The released DNA was then placed into the extended slots of a Minislot 30 apparatus (Immunetics; Cambridge, MA, USA), concentrated on a 15 × 15 cm positively charged nylon membrane (Boehringer Mannheim; Indianapolis, IN, USA), and fixed to the membrane by baking it at 120 °C for 20 min. The membrane was placed in a Miniblotter 45 (Immunetics; Cambridge, MA, USA) with the lanes of DNA at 90° to the lanes of the device. Digoxigenin-labelled whole genomic DNA probes for 40 bacterial species were hybridized in individual lanes of a Miniblotter 45. After hybridization, the membranes were washed at high stringency, and the DNA probes were detected using the antibody to digoxigenin conjugated with alkaline phosphatase and chemiluminescence detection. The last two lanes in each run contained standards at concentrations of 10^5^ and 10^6^ cells of each species. Signals were converted to absolute counts by comparison with the standard lanes on the membrane. The sensitivity of the assay was adjusted to permit the detection of 10^4^ cells of a given species by adjusting the concentration of each DNA probe. This procedure was previously described by Mestnik et al. (2010) [6].

### 4.5. Statistical Analysis

The mean counts (×10^5^) of individual bacterial species were averaged within each subject, then across subjects in both groups. The percentage of the total DNA probe counts was determined initially in each site, then per subject, and averaged across subjects in the four groups. Adjustments for multiple comparisons were performed when the 40 bacterial species were evaluated simultaneously [38,39,40]. The significance of differences among the groups was sought using Kruskal–Wallis and Mann–Whitney tests. Friedman and Dunn’s multiple comparison tests were used to test differences among time points. The Statistical Package for the Social Sciences (SPSS 22, 2013 software, New York, NY, USA) was used for statistical analysis, and the level of significance was set at 5%.

## 5. Conclusions

Taking into account the limits of this study, it can be concluded that:−The concomitant use of antimicrobial chemical control (systemic and local) associated with basic periodontal therapy appears to have a considerable effect on the microbial composition of the oral cavity, especially in the subgingival biofilm and saliva;−The extracrevicular sites presented different microbial patterns compared with the intracrevicular site (subgingival plaque);−The impact of combined antimicrobial chemical control reduced or minimized the subgingival recolonization around remaining periodontal pockets.

## Figures and Tables

**Figure 1 antibiotics-12-00847-f001:**
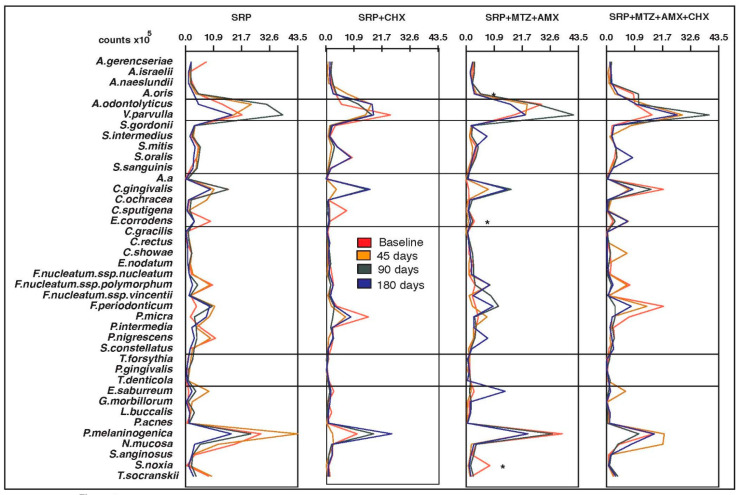
Mean counts (×10^5^) of the 40 test species in samples from tongue coating during the experimental period. SRP, scaling and root planing; CHX, chlorhexidine; MTZ, metronidazole; AMX, amoxicillin. * *p* < 0.05, Friedman test: significance of differences among the time points within each group.

**Figure 2 antibiotics-12-00847-f002:**
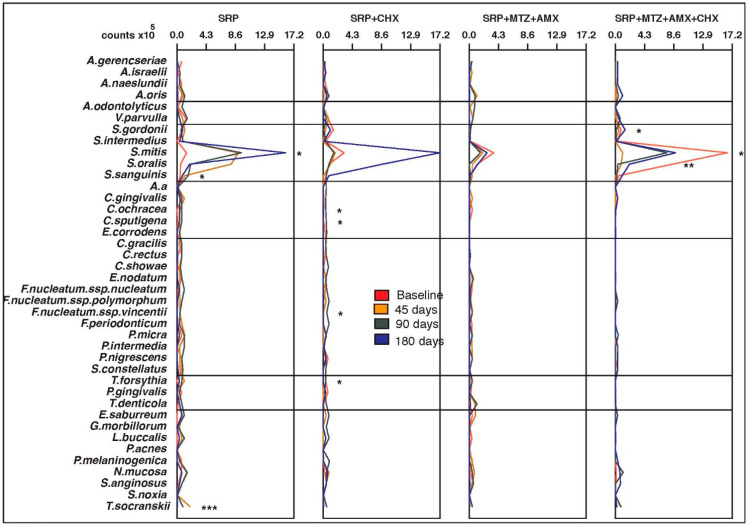
Mean counts (×10^5^) of the 40 test species in samples from soft tissue during the experimental period. SRP, scaling and root planing; CHX, chlorhexidine; MTZ, metronidazole; AMX, amoxicillin. * *p* < 0.05; ** *p* < 0.01; *** *p* < 0.001, Friedman test: significance of differences among the time points within each group.

**Figure 3 antibiotics-12-00847-f003:**
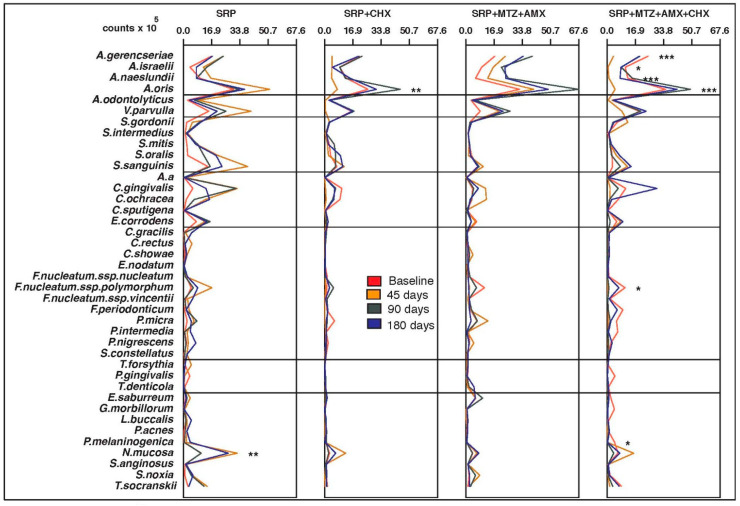
Mean counts (×10^5^) of the 40 test species in samples from supragingival biofilm during the experimental period. SRP, scaling and root planing; CHX, chlorhexidine; MTZ, metronidazole; AMX, amoxicillin. * *p* < 0.05, ** *p* < 0.01, *** *p* < 0.001, Friedman test: significance of differences among the time points within each group.

**Figure 4 antibiotics-12-00847-f004:**
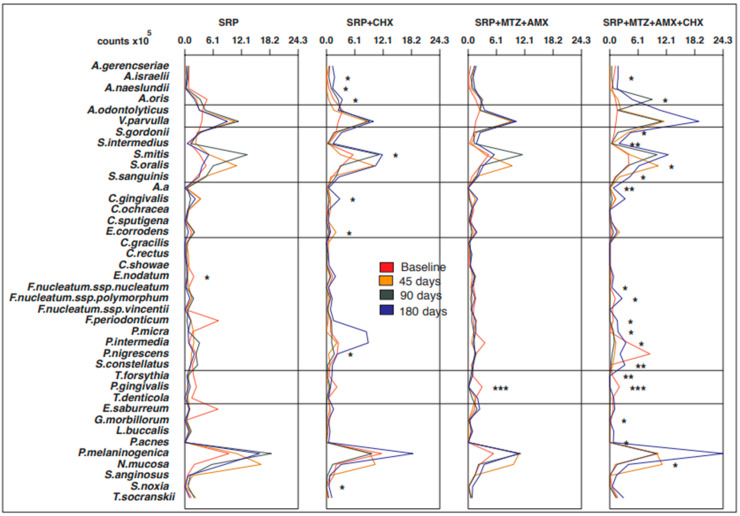
Mean counts (×10^5^) of the 40 test species in samples from saliva during the experimental period. SRP, scaling and root planing; CHX, chlorhexidine; MTZ, metronidazole; AMX, amoxicillin. * *p* < 0.05; ** *p* < 0.01; *** *p* < 0.001, Friedman test: significance of differences among the time points within each group.

**Figure 5 antibiotics-12-00847-f005:**
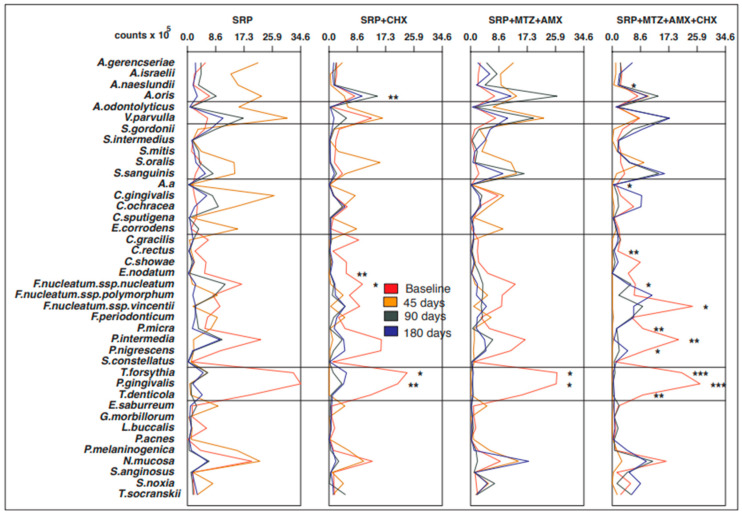
Mean counts (×10^5^) of the 40 test species in samples from subgingival biofilm during the experimental period. SRP, scaling and root planing; CHX, chlorhexidine; MTZ, metronidazole; AMX, amoxicillin. * *p* < 0.05; ** *p* < 0.01; *** *p* < 0.001, Friedman test: significance of differences among the time points within each group.

**Figure 6 antibiotics-12-00847-f006:**
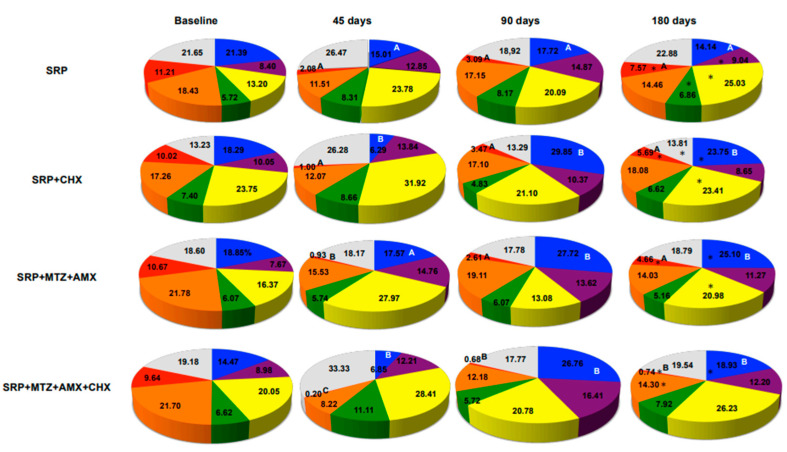
Pie charts of the mean proportion of each microbial complex in samples from the five intraoral niches (dorsum of the tongue, soft tissue, saliva, supragingival, and subgingival biofilms) analyzed together during the experimental period. SRP, scaling and root planing; CHX, chlorhexidine; MTZ, metronidazole; AMX, amoxicillin. * *p* < 0.01, Dunn’s multiple comparison tests: significance of differences within each group between baseline and 180 days post therapy. Kruskal–Wallis and Mann–Whitney test: different letters indicate statistically significant differences among groups at each time point.

**Table 1 antibiotics-12-00847-t001:** Demographic characteristics and mean ± SD full-mouth clinical parameters at baseline and follow-up visits.

Variable	Time Point	Treatments Groups				*p*-Value
		SRP N = 15	SRP + CHX N = 15	SRP + MTZ + AMX N = 15	SRP + MTZ + AMX + CHX N = 15	
Gender (male/female)	Baseline	03/12	05/10	06/09	05/10	0.412 ^#^
Age	Baseline	45.81 ± 8.54	43.44 ± 8.26	46.37 ± 8.59	44.95 ± 8.77	0.223 *
PD (mm)	Baseline	3.72 ± 0.71	3.85 ± 0.74	3.88 ± 0.69	3.83 ± 0.78	0.541 *
CAL (mm)	Baseline	4.22 ± 0.93	4.34 ± 0.96	4.23 ± 1.08	4.24 ± 0.98	0.418 *
Mean number of sites with PD ≥ 5 mm	Baseline	40.78 ± 20.67	40.12 ± 19.66	38.31 ± 22.16	39.73 ± 21.74	0.479 *
	180 days	18.11 ± 13.62 ^a^	19.56 ± 12.49 ^a^	4.33 ± 5.01 ^b^	3.91 ± 5.87 ^b^	0.0001 *
Δ in the number of sites with PD ≥ 5 mm	-	22.67 ± 2.98 ^a^	21.55 ± 1.17 ^a^	33.98 ± 1.71 ^b^	35.82 ± 1.93 ^b^	0.0001 **

Values are presented as means ± standard deviations (SDs). The significance of differences among groups for the proportion of males was assessed using the chi-square test (#). The significance of differences among groups for age and baseline clinical parameters was evaluated using a one-way ANOVA (*). The significance of differences among groups for mean change in the number of sites with PD ≥ 5 mm from baseline to 180 days was calculated using an ANCOVA (**), adjusting for the baseline numbers of sites with PD ≥ 5 mm. Mann–Whitney test: different letters indicate statistically significant differences among groups at each time point. SRP; scaling and root planing; MTZ, metronidazole; AMX, amoxicillin; PD, probing depth; CAL, clinical attachment level; CHX, chlorhexidine.

## Data Availability

Not applicable.

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
