# Peer review of "Systemic Antibiotics and Chlorhexidine Associated with Periodontal Therapy: Microbiological Effect on Intraoral Surfaces and Saliva"

_antibiotics, 2023, doi:10.3390/antibiotics12050847_

Round 1

Reviewer 1 Report

This article which highlights the use of MTZ and AMX with chlorhexidine associated with periodontal therapy and the microbiological effect on intra-oral surfaces and saliva is a very complete and interesting study for the scientific community and of course for further use. However, the choice of this mixture of products and of amoxicillin as an antibiotic does not seem, to me, to be sufficiently explained in the introduction and even in the discussion. “MTZ and AMX appears to be the most favorable systemic antibiotic used in the treatment of periodontitis” this statement is referenced but could the authors give two or three arguments as to why?

An additional paragraph on the use of this broad-spectrum antibiotic rather than another should be added either in the introduction or in the discussion. Indeed it is mentioned the study of several antibiotics and six systematic reviews on the topic but a small summary of the results and therefore of the choice of amoxicillin and its advantages over other broad-spectrum antibiotics would be appreciated.

Otherwise the study is important, interesting and the results are well explained, so this article deserves to be published after making this addition.

Author Response

Reviewer#1

The concern of the Reviewer: This article which highlights the use of MTZ and AMX with chlorhexidine associated with periodontal therapy and the microbiological effect on intra-oral surfaces and saliva is a very complete and interesting study for the scientific community and of course for further use.

Answer: Thanks for your comments.

Revised text: N.A

The concern of the Reviewer: However, the choice of this mixture of products and of amoxicillin as an antibiotic does not seem, to me, to be sufficiently explained in the introduction and even in the discussion. “MTZ and AMX appears to be the most favorable systemic antibiotic used in the treatment of periodontitis” this statement is referenced but could the authors give two or three arguments as to why?

Answer: Thanks for raising this critical point. We added more details about the rationality of the use of combined systemic antibiotics for the treatment of periodontal diseases as well as the impact of this association on the clinical outcomes.

Revised text: The association of systemic and local chemical controls of dental biofilm with SRP has been suggested as an adjunct treatment for periodontitis. In this context, metronidazole (MTZ) combined with amoxicillin (AMX) appears to be the most favorable systemic antibiotic used in the treatment of periodontitis [4-8]. MTZ is a synthetic drug derived from nitroimidazole and presents bacterial and protozoan activity. MTZ acts on the bacterial cells by passive diffusion and provides toxic metabolites that interact with DNA and other bacterial macromolecules, causing cellular death [9]. However, this effect is limited to strict anaerobic bacteria, such as Porphyromonas gingivalis, Treponema denticola, and Tanerella forsythia. AMX is a bactericidal antibiotic from the penicillin group, acting on anaerobic facultative bacteria, coccus, and bacillus Gram-negative and Gram-positive. AMX avoids the synthesis of the bacterial cell wall resulting in bacterial death.

An earlier review [10] showed that scaling and root planning (SRP) associated with systemic MTZ+AMX reduced deeper pockets >5mm more effectively than SRP alone. The broad rebiosis in the subgingival bacterial plaque goes beyond its effects in controlling species such as Aggregatibacter actinomycetemcomitans. MTZ+AMX acted on the reduction of several periodontal pathogens from the red and orange complexes and some newly identified taxa and an increase in the proportions of host-compatible species [6-8, 11, 12]. In addition, combining chlorhexidine (CHX) mouth rinse with SRP leads to clinical benefits and can improve the reduction of periodontal pathogens from different intraoral surfaces and saliva [8, 11, 12].

The concern of the Reviewer: An additional paragraph on the use of this broad-spectrum antibiotic rather than should be added either in the introduction or in the discussion. Indeed it is mentioned the study of several antibiotics and six systematic reviews on the topic but a small summary of the results and therefore of the choice of amoxicillin and its advantages over other broad-spectrum antibiotics would be appreciated.

Answer: Thanks for your comment. This concern was also described in the same sentence above.

Revised text: MTZ is a synthetic drug derived from nitroimidazole and presents bacterial and protozoan activity. MTZ acts on the bacterial cells by passive diffusion and provides toxic metabolites that interact with DNA and other bacterial macromolecules, causing cellular death [9]. However, this effect is limited to strict anaerobic bacteria, such as Porphyromonas gingivalis, Treponema denticola, and Tanerella forsythia. AMX is a bactericidal antibiotic from the penicillin group, acting on anaerobic, facultative bacteria, coccus, and bacillus Gram-negative and Gram-positive. AMX avoids the synthesis of the bacterial cell wall resulting in bacterial death.

An earlier review [10] showed that scaling and root planning (SRP) associated with systemic MTZ+AMX reduced deeper pockets >5mm more effectively than SRP alone. The broad rebiosis in the subgingival bacterial plaque goes beyond its effects in controlling species such as Aggregatibacter actinomycetemcomitans. MTZ+AMX acted on the reduction of several periodontal pathogens from the red and orange complexes and some newly identified taxa and an increase in the proportions of host-compatible species [6-8, 11, 12]

The concern of the Reviewer: Otherwise the study is important, interesting and the results are well explained, so this article deserves to be published after making this addition.

Answer: Thank you!

Revised text: N.A

Reviewer 2 Report

Dear Authors,

Thank you for submitting this manuscript. I think the paper is quite interesting because it refers to a very important topic: the evaluation of the microbiological effects on different oral cavity sites of scaling and root planning (SRP) combined with antimicrobial chemical control in the treatment of periodontitis. I would like to suggest some points to the Authors:

1. The abstract should include a short statement on the current research gap and question to show why this study is unique and worthy of publication.

2. The introduction is too short, please add more information about SRP and the different clinical methods of periodontal treatment and why is this significant for your study.

3. In the Introduction section, line 44: Please add more references.

4. Lines 39 – 42: This sentence is too long, please revise it.

5. You should follow the instructions for the authors – the section Materials and methods should be after the Introduction section.

6. The authors should add the null and working hypotheses of the study and highlight them by adding "H0" and "H1"

7. The description of figure 6 is too long, please reduce it to one short sentence or separate it into several figures.

8. Please provide the informed consent statement form.

9. In the Conclusion section, please describe the significance of this study. The authors should summarize the significant findings in bullets for clarity in the Conclusion section.

Thank you in advance for all the corrections. Good luck!

Author Response

REVIEWER#2

The concern of the Reviewer: The abstract should include a short statement on the current research gap and question to show why this study is unique and worthy of publication.

Answer: Well pointed. The abstract was revised accordingly.

Revised text: “The effect of systemic antibiotics on the microbial profile of extra-crevicular sites after periodontal treatment is currently under research. This study evaluated the microbiological effects on different oral cavity sites of scaling and root planing (SRP)…”

The concern of the Reviewer: The introduction is too short, please add more information about SRP and the different clinical methods of periodontal treatment and why this is significant for your study.

Answer: Thanks for raising this important point. We added more details about the rationality of the use of combined systemic antibiotics for the treatment of periodontal diseases as well as the impact of this association on the clinical outcomes.

Revised text: The association of systemic and local chemical controls of dental biofilm with SRP has been suggested as an adjunct treatment for periodontitis. In this context, metronidazole (MTZ) combined with amoxicillin (AMX) appears to be the most favorable systemic antibiotic used in the treatment of periodontitis [4-8]. MTZ is a synthetic drug derived from nitroimidazole and presents bacterial and protozoan activity. MTZ acts on the bacterial cells by passive diffusion and provides toxic metabolites that interact with DNA and other bacterial macromolecules, causing cellular death [9]. However, this effect is limited to strict anaerobic bacteria, such as Porphyromonas gingivalis, Treponema denticola, and Tanerella forsythia. AMX is a bactericidal antibiotic from the penicillin group, acting on anaerobic facultative bacteria, coccus, and bacillus Gram-negative and Gram-positive. AMX avoids the synthesis of the bacterial cell wall resulting in bacterial death.

An earlier review [10] showed that scaling and root planning (SRP) associated with systemic MTZ+AMX reduced deeper pockets >5mm more effectively than SRP alone. The broad rebiosis in the subgingival bacterial plaque goes beyond its effects in controlling species such as Aggregatibacter actinomycetemcomitans. MTZ+AMX acted on the reduction of several periodontal pathogens from the red and orange complexes and some newly identified taxa and an increase in the proportions of host-compatible species [6-8, 11, 12]. In addition, combining chlorhexidine (CHX) mouth rinse with SRP leads to clinical benefits and can improve the reduction of periodontal pathogens from different intraoral surfaces and saliva [8, 11, 12].

The concern of the Reviewer: In the Introduction section, line 44: Please add more references.

Answer: The manuscript was revised as suggested.

Revised Text: Additionally, combining chlorhexidine (CHX) mouth rinse with SRP leads to clinical benefits and can improve the reduction of periodontal pathogens from different intraoral surfaces and saliva [8-14].

The concern of the Reviewer: Lines 39 – 42: This sentence is too long, please revise it.

Answer: Thanks for your comment. The manuscript was revised accordingly.

Revised Text: The clinical success of periodontal treatment is achieved when the levels, proportions, and percentages of sites colonized by different periodontal pathogens are effectively reduced after therapy. Complementary, the successful outcome also showed a new microbial community with higher proportions of host-compatible microorganisms are established in the subgingival biofilm [13-15].

The concern of the Reviewer: You should follow the instructions for the authors – the section Materials and Methods should be after the Introduction section.

Answer: The manuscript followed the instructions of the authors (see https://www.mdpi.com/journal/antibiotics/instructions#manuscript). In this link, the MDPI editor clearly states that: results must be described after the introduction section for the paper from the research manuscript sections.

Revised text: N.A

The concern of the Reviewer: The authors should add the null and working hypotheses of the study and highlight them by adding "H0" and "H1"

Answer: Well pointed. We added the hypothesis in the revised paper.

Revised text: To our knowledge, no other study has evaluated the changes in the microbial profile of soft tissues and saliva after periodontal treatment with systemic and local antimicrobials in subjects with periodontitis. Thus, this study evaluated the hypothesis that SRP combined with systemic MTZ and AMX associated with CHX mouth rinse affected (H1) or not (H0) the microbiological composition of different oral cavity sites in the treatment of periodontitis.

The concern of the Reviewer: The description of figure 6 is too long, please reduce it to one short sentence or separate it into several figures.

Answer: Thanks for your comment. However, this figure aimed to compare the microbiological patterns in different sites at the same time. Therefore, we reduce the legend of the figure to facilitate the reader's comprehension. 

Revised Text: Figure 6. Pie charts of the mean proportion of microbial complex from the dorsum of the tongue, soft tissue, saliva, supragingival and subgingival biofilms. SRP, scaling and root planing; CHX, chlorxedine; MTZ, metronidazole; AMX, amoxicillin.*p<0.01, Dunn´s multiple comparison tests: significance of differences within each group between baseline and 180 days post-therapy. Kruskal-Wallis and Mann-Whitney test (different letters indicate statistically significant differences among groups at each time point).

The concern of the Reviewer: Please provide the informed consent statement form.

Answer: Please, find attached, at the end of this section, the copies of the IRB approval of the study and the forms.

The concern of the Reviewer: In the Conclusion section, please describe the significance of this study. The authors should summarize the significant findings in bullets for clarity in the Conclusion section.

Answer: Thanks for your comment. The manuscript was revised accordingly.

Revised Text: Conclusions

Within the limits of this study, it can be concluded that:

- The concomitant use of antimicrobial chemical control (systemic and local) associated with basic periodontal therapy appears to have a deeper effect on the microbial composition of the oral cavity, especially in the subgingival biofilm and saliva

- The extra-crevicular sites presented different microbial patterns when compared with the intra-crevicular site (subgingival plaque)

- The impact of combined antimicrobial chemical control could reduce or minimize the subgingival recolonization around remaining periodontal pockets.